# Machine Learning Approaches for Metalloproteins

**DOI:** 10.3390/molecules27041277

**Published:** 2022-02-14

**Authors:** Yue Yu, Ruobing Wang, Ruijie D. Teo

**Affiliations:** 1Division of Natural and Applied Sciences, Duke Kunshan University, Kunshan, Jiangsu 215316, China; yy241@duke.edu; 2Department of Physics, Duke University, Durham, NC 27708, USA; 3Department of Chemistry, Duke University, Durham, NC 27708, USA; ruobing.wang@duke.edu; 4UNC Eshelman School of Pharmacy, University of North Carolina at Chapel Hill, Chapel Hill, NC 27599, USA

**Keywords:** metalloproteins, metalloenzymes, machine learning, deep learning, protein structure, protein function, protein stability, inhibitor design, cleavage sites

## Abstract

Metalloproteins are a family of proteins characterized by metal ion binding, whereby the presence of these ions confers key catalytic and ligand-binding properties. Due to their ubiquity among biological systems, researchers have made immense efforts to predict the structural and functional roles of metalloproteins. Ultimately, having a comprehensive understanding of metalloproteins will lead to tangible applications, such as designing potent inhibitors in drug discovery. Recently, there has been an acceleration in the number of studies applying machine learning to predict metalloprotein properties, primarily driven by the advent of more sophisticated machine learning algorithms. This review covers how machine learning tools have consolidated and expanded our comprehension of various aspects of metalloproteins (structure, function, stability, ligand-binding interactions, and inhibitors). Future avenues of exploration are also discussed.

## 1. Introduction

When oxygen molecules enter the human body through the lungs, they attach to hemoglobin molecules in red blood cells by oxidizing the Fe^2+^ heme cofactor, producing a low-spin, ferric-oxy complex. Should the iron in heme be rendered defunct, our bodies would lack the most fundamental component for aerobic respiration. Proteins that contain metal cofactors, such as hemoglobin, are termed metalloproteins, and they make up nearly half of the entire protein population [1]. Aside from hemoglobin, another example of a critical metalloprotein is ceruloplasmin [2], which prevents Fe^2+^ from being oxidized prematurely in the bloodstream. Furthermore, alcohol dehydrogenase, the enzyme responsible for breaking down toxic alcohol in humans and other animals, relies on a zinc ion to coordinate its substrate [3]. Other organisms also take advantage of metalloproteins, most notably every green plant species with proteins containing chlorophyll, which binds to Mg^2+^. Magnesium provides structural integrity to chlorophyll, the molecule responsible for harnessing energy for the entire biosphere by facilitating the efficient capture and transfer of energy from antenna pigments (i.e., chlorophyll b) to the reaction center.

The abundance and ubiquity of metalloproteins have attracted much scientific endeavors to understand the relationships between protein structures and functions [4,5], and translate that understanding into real-life applications [6,7]. While traditional molecular modeling approaches, such as classical molecular dynamics [8,9] and quantum mechanics/molecular mechanics (QM/MM) methods [10], have often been used to study these objectives, the usage of machine learning models has grown in popularity over the last decade [11], as metalloproteins can now be studied in a computationally inexpensive manner at a systems level. By designing and optimizing models to learn patterns and distinctions from training and validation data sets, these machine learning models can eventually predict the properties and behaviors of any new inputs (i.e., test sets). In other words, the machine “learns“ to make judgments about its input. The ability of the model to learn patterns from large datasets [12] makes it advantageous yet complementary to experimental and the aforementioned molecular modeling approaches [13]. Machine learning can be divided into three main types: supervised learning (uses labeled data for new output predictions), unsupervised learning (uses unlabeled data to identify patterns), and reinforcement learning (uses a learning agent to predict the correct output by maximizing its reward). Some common examples of supervised learning algorithms include random forests [14], support vector machines (SVM) [15], and linear regression. Unsupervised learning is used for clustering and association (i.e., using hidden Markov models (HMMs) [16]). A neural network, on the other hand, is a popular type of architecture that can be used for both supervised and unsupervised learning. It models the human brain, where information flows across axons of various synaptic weights, and different sets of data features are evaluated at various points (using corresponding channels), much like a neural transmission [17]. In addition, neural networks often contain many hidden layers of interconnected neurons to model more complex problems (more commonly known as deep learning).

Machine (and deep) learning has recently displayed great potential in studying protein structures. In 2018, the first version of AlphaFold, a protein structure prediction algorithm based on deep convolutional residual neural networks, was developed by Google AI’s DeepMind [18,19] and outperformed all other programs for the thirteenth edition of the critical assessment of protein structure prediction (CASP). Two years later, during the fourteenth edition of CASP, the second version of AlphaFold, which incorporates the revolutionary attention mechanism [20], made virtually impeccable predictions for various amino acid sequences [13] and predicted structures corresponding to 98.5% of the entire human proteome [21]. The results were highlighted by *Nature* [22], and it was hailed as a milestone in deep learning applications and biology. However, as famous as this accomplishment is, machine learning has found its way into studying proteins long before it. For years, researchers have developed programs that could analyze the functionality of enzymatic sites by finding the sequence and structural patterns that could predict what a protein might do and what pathology could ensue from defects at specific loci. The purpose of these endeavors is that, eventually, integrating complex biomolecular theories into the practical design of functional proteins becomes feasible computationally. Ideally, algorithms will determine the protein structure from an amino acid sequence, and combined with experimental validation, one can recreate naturally-occurring proteins and tailor *de novo* proteins to fulfill specific objectives.

In this review, our focus will be on metalloproteins, of which there has been much data relating to structure and properties gathered over time. The large amount of experimental data makes metalloproteins the perfect subject for machine learning approaches. Using various algorithms (random forest, neural network, SVM, etc.) [23], not only can one study how metalloproteins structure themselves with the metal they bind, but also how that relates to functionality, and how one can, in theory, design new proteins. Hence, the focus of this review is distinct from other reviews on non-metal-containing proteins; for example, this review covers unique challenges that machine learning can resolve (see Section 4), such as differentiating between catalytic metal sites and inactive sites that are structurally similar [12]. Furthermore, this review summarizes studies related to other properties, such as metalloprotein stability, inhibition, and how and where substrate cleaving occurs.

## 2. Structural Analysis

In the past decades, there have been multiple attempts to understand metalloproteins using theoretical or computational methods, while many established databases and software documents have analyzed metal-binding processes [24,25,26]. For example, in 2000, Dudev and Lim [27] conducted ab initio and continuum dielectric calculations of the free energy change that occurs when a protein binds a metal ion in the presence of surrounding water molecules. The study aimed to understand why ions bind to hydrophilic residues directly instead of first coordinating water ligands. The results show that a low dielectric constant, which depends on the metal’s chemical environment, increases the binding affinity of the ion to the hydrophilic core. Computationally, Dudev et al. [28] surveyed the protein data bank (PDB) for protein structures containing Zn, Ca, Mn, and Mg to investigate the influence of the second shell on metal binding and selectivity. Then, in 2012, Andreini et al. [29] presented the FindGeo program, which was used to determine the geometry of metal ion coordination. Written using FORTRAN 77 and Python, it takes PDB files as input and finds ion-coordinating amino acid residues, determined using atoms within a specified threshold distance. The method proves to be effective for determining less regular metalloprotein geometries. While these endeavors have done much to deepen our understanding of metalloproteins, it remains at the molecular level of obtaining new knowledge relevant to specific proteins. In order to broaden the applicability of metalloprotein research, one should utilize a systems-level approach to engender new and testable knowledge by exploiting the vast collection of data. In 1999, DeGrado et al. [30] published an article on how to characterize and design novel metalloprotein structures computationally. Although some plausible clues about constructing new structures were illuminated, the observations amounted to generalizations. A more straightforward approach would be to develop an algorithm that takes any amino acid sequence and predicts what behavior it will possess (or not) as a folded protein.

This approach has become possible with the advent of machine learning and deep learning [31]. AlphaFold has been a hallmark success, but the application of machine learning in protein studies was already underway before it. For metalloproteins alone, in 2007, Passerini et al. [32] applied an SVM to predict the existence of zinc metal-binding sites within the human proteome. The results were promising, as the model accurately predicts metal binding for some residues, and many of its predictions are confirmed by previous works. It was significantly fallible, however, as its predictions are not assured to be valid for metalloproteins *in vivo*: a protein it predicts to bind zinc might bind to other metals. Four years later, Passerini et al., released MetalDetector v2.0 [33], which uses a combination of algorithms (SVM, HMM, recurrent neural network) to predict metal binding sites in proteins, including those without known structural similarities. Following this study, another metal-binding site predictor, DeepMBS, was published [34]; it is based on a deep convolutional neural network and is the first application of a deep learning structure to such a site prediction. In 2014, Estellon et al. [35] published a study on identifying microbial iron-sulfur proteins using a HMM. After experimental validation, two new proteins containing iron–sulfur clusters were identified. The examples above were undoubtedly innovative at that time. However, a heavy setback is the limited scope of what these models target.

The issue of scope did not go unattended, however. In 2005, Lin et al. [36] presented an artificial neural network that could predict metal-binding residues in a metalloprotein based on its primary sequence. By limiting themselves to strictly biologically-relevant features (i.e., solvent-exposed surface area, secondary structure, amino acid physical properties, chemical properties, hydrophobicity) and using data from the PDB and the metalloprotein database, the authors designed a feedforward neural network. More than 90% specificity in the predictions was achieved. For the metal elements tested, bulk and trace metal predictions yielded >98% accuracy, and the application of one particular feature set (corresponding to the amino acid chemical properties) displayed an impeccable sensitivity of almost 100% for bulk metals. In 2008, Carugo published a study [37] along a similar line of logic. The goal was to design a versatile program that determined if a protein required a metal ion and whether it could discern which species of metal it required. Drawing from the UniProt database, it took as variables the frequency of various amino acid cluster structures, which are distinctive based on their conformations, hydrophobicities, and whether they fold. The program itself uses a random forest, and the author discussed its sensitivity, specificity, precision, and accuracy when predicting whether a protein could incorporate a specific metal type or not. Although the results show that the model has varying accuracies, depending on the type of metal, all of the performance metrics indicate that the model performs better than a random classifier.

These works show the progress in machine learning applications towards versatility and prediction reliability, long before the method had primarily captured the public’s attention. Several years later, machine learning approaches were used to provide more holistic answers to incompletely-answered questions. For instance, Liu and Altman developed an enhanced version of the FEATURE program [38,39] by coupling loop modeling with a Naïve Bayes (NB) classifier trained using features related to biochemical and structural properties, which could predict calcium-binding sites in disordered regions with 70% accuracy, thus expanding our repertoire of metal-binding site predictors to putative conformations and regions that cannot be crystallized. On a related note, neural networks have also been applied to the development of molecular potentials and force fields for molecular dynamics simulations of metalloproteins [40,41,42], which in turn, can be used for structural refinement and investigating other relevant processes, such as ligand binding. In another study, Brylinski and Skolnick [43] addressed the above issue of metal binding in a 2012 study, where a new program called FINDSITE-metal was presented. This algorithm combines structural and evolutionary information (using templates with varying sequence identity to the target) with an SVM trained to assign a binding probability between a protein residue and a metal ligand. As with Passerini et al. [32], the focus involves predicting metal-binding proteins within the human proteome. One distinction of this research is that the authors found that evolutionary-related proteins bind similar metals at the exact locations with identical residues. Another key finding of the FINDSITE-metal study is the dependency of the percentage of correctly predicted sites on native and distorted structures. Through a Monte Carlo process, Brylinski and Skolnick first generated distortions to known crystal structures (at specific resolutions). The fraction of correctly predicted sites (defined to be within a 4 Å distance between the predicted metal location and metal location of the target structure) decreased slightly from 69.5% (crystal structure) to 67.2% for a 2 Å RMSD-distorted structure and as low as 50.8% for a 6 Å RMSD-distorted structure. When the distance between the crystal structure- and template-bound metal ion increases to more than 4 Å, the predicted protein tends to bind to non-native ions instead of native ions. This observation also holds for the distorted structures. This multi-faceted work of Brylinski and Skolnick takes an early step into using machine learning to engineer artificial metalloproteins, bringing us closer to turning our knowledge towards actual usage. In recent years, researchers have made this goal explicit, presenting how machine learning can boost efforts to engineer new structures.

## 3. Structural Design

The rules for selecting appropriate metal-coordinating residues and designing sites of high metal-binding affinity, as well as the factors that determine ion selectivity at these sites, have been investigated to a large extent by the *de novo* protein design community [30,44]. Nevertheless, manually navigating through these rules is highly impractical, even for recreating a known structure, let alone designing an altogether new one. For such a reason, many regarded computation as a viable alternative. As long as the predictions are reliable, the process to achieve the desired protein structure is accelerated when experimentalists can validate high-confidence predictions. In turn, new experimental data becomes available to expand training sets and update machine learning models, such that there is constant synergy between both computation and experimentation (otherwise known as active learning). The need to experimentally validate structure predictions was one of the topics in Lu’s discourse [45] on metalloprotein design, which discussed limitations related to binding site geometries and the lack of selectivity among different ions. As recent as 2021, Osadchy and Kolodny [46] mapped out a theory on how to build a deep learning network that would produce sequences resembling natural proteins for desired functions. It was postulated that by studying copious amount of structures, it would be possible to teach various generative models (autoregressive, energy-based, variational autoencoders, normalizing flow, generative adversarial networks) to predict an input amino acid sequence with a predefined property. Subsequently, other computational methods, such as Rosetta and molecular dynamics, can evaluate these predicted properties, although experimental validation is unequivocally the gold standard.

In 2018, Greener et al. [47] explored the combination of protein design and redesign with deep generative models. In the study, conditional variational autoencoders (CVAE), an inference-generation mechanism capable of outputting protein sequences that match certain attributes (such as metal-binding or topological conformation), were used. With this ability, the objective was to use unsupervised learning to design metal-binding sites in non-metalloproteins and to write out amino acid sequences for brand new topologies (Figure 1). The authors leveraged molecular dynamics to compute a minimal energy structure according to the output sequence for the latter purpose. As for structural design, modifications to sites in natural non-metal-binding human proteins were made to bind metals by changing a small number of amino acids. Although no experimental validation was yet known, this result indicates that the information about metalloproteins can be expanded and new metalloproteins can be designed according to specific needs. Given the ubiquity and multi-functionality of metalloproteins [48], this capability could affect how human diseases arising from the loss of metalloprotein activity are treated. If the same is possible for defective metal-binding proteins by showing where the defects arise in their sequences, this approach can evolve into a novel and effective diagnostic/therapeutic method.

Incidentally, the diagnostic aspect is precisely what Koohi-Moghadam et al. [49] had in mind in a study on disease-related mutations using deep learning [50]. In order to study the diverse types of human disorders caused by missense mutations in metalloproteins, sequential and spatial configuration data of multiple distinct disease-related mutations, along with several benign ones, were extracted. This dataset was used to train a multichannel convolutional neural network (MCCNN) that would ideally predict if a site is associated with a disease for proteins that bind any metal type, though their examples were limited only to Zn, Ca, and Mg due to sampling size limitations. To demonstrate MCCNN’s effectiveness, the authors compared its performance with PolyPhen-2, a NB classifier algorithm designed to predict amino acid substitution effects. Ultimately, MCCNN excels for all performance metrics except for sensitivity (Table 1 of reference [49]), which was attributed to a dearth of training data. In short, with more extensive training, future deep learning networks could very likely produce much more diverse and reliable disease predictions. Mutations are nature’s redesign of existing proteins, albeit more often for the worse. Learning from nature’s mistakes can assist us in avoiding them in our endeavor to make functionally-meaningful proteins and even reverse the consequences should there be a deviation.

## 4. Function

As with all proteins, the metalloprotein structure determines its function, and the ultimate goal of structural alteration or design is to achieve wanted functions. It is noteworthy that function is relevant to what substrate a protein binds, what happens to the substrate during binding, and inherent physical properties of the protein, such as resistance to external stress. In 2011, Chellapandi [51] reviewed several cases of metalloenzyme design using various computational methods, including machine learning. The discussion contained many software algorithms and QM/MM methods built for constructing new enzymes that perform naturally occurring functions. These methods were able to design enzymes such as phosphate-dependent aldolases that are comparable in activities to their natural counterparts, although controlling the stereoselectivity remains a challenge [52]. However, several more competitive methods, such as amino acid or metal ion replacement and machine learning, have arisen [53].

Concerning machine learning, in 2007, Liao et al., presented a study on engineering proteinase K employing this approach [53]. Eight different algorithms (ridge regression (RR), least absolute shrinkage, and selection operator (Lasso), partial least square regression (PLSR), support vector machine regression (SVMR), linear programming support vector machine regression (LPSVMR), linear programming boosting regression (LPBoostR), matching loss regression (MR), one-norm regularization matching-loss regression (ORMR)) that differed in the discrepancy between predicted and actual enzyme activity, and what regularization function (L1-norm vs. L2-norm) was applied, were assayed. The authors could synthesize proteinase K variants based on any desired amino acid sequence, intending to produce functional enzymes that display resistance to overheating. Three rounds of designing were run, each round building upon the previous to enhance enzymatic activity. In the end, after synthesizing and testing 95 select variants, proteinase activity was increased by a factor of 20. The authors predict that this strategy is transferable to modifying other proteins, reducing the need for large amounts of variants, and disposing the need for protein libraries and massive screening.

As machine learning algorithms have advanced in recent years, it is no surprise that more works on protein function prediction have emerged. In 2019, Zou et al. [54] published an article on a deep learning program, mlDEEPre, specialized to foretell what multi-functional proteins can do, which turned out to possess unquestionably high reliability by all measures and was an overall improvement to multiple previous models. It did not, however, specifically address metalloenzymes. Later studies have made that focus, including a study by Soni et al. [55], which aimed to enhance the predictability of protein-ligand binding affinities (for both metallo- and non-metallo complexes). Their program, Bappl+, scores the protein-ligand binding affinity by taking into account interaction energies and entropies before passing it into training random forests. After evaluating the performance of Bappl+ against three test datasets of protein-ligand complexes, it displayed superiority compared to most existing scoring functions (see Table 2 of reference [55]).

In 2021, Feehan et al. [12] took the largest structure database of enzymatic/non-enzymatic metalloproteins to train a decision-tree ensemble to differentiate between these two categories. Numerous models were examined in search of the one with optimal performance using test sets (Figure 2), using the Matthews correlation coefficient (MCC) to prevent bias due to imbalance in the training set (76% non-enzymatic data). MAHOMES, the decision-tree ensemble marked on the upper-rightmost corner in Figure 2 and optimized for precision and MCC, with predictions based on metal-binding sites, was evaluated and compared to other enzymatic/non-enzymatic predictors. With a 92.2% precision, MAHOMES outmatches sequence-based models such as DEEPre, EFICAz2.5, and DeepEC, as well as catalytic residue-based ones, such as CRPred, CRHunter, and PreVAIL. Its accuracy surpasses sequence-based but falls behind residue-based, and vice versa for recall, so its superiority is dependent on context.

Furthermore, Feehan et al., were able to determine which features of metalloproteins, such as electrostatics and binding pocket geometries, were most representative of its enzymatic nature. Using a Jaccard index for various features, a scale was developed where 0 means that the features are entirely the same between enzymatic/non-enzymatic and 1 means the opposite. In this case, the Rosetta energy summed over the spherical volume of the binding site turns out to be distinctive. The role of volume as a key player becomes pertinent considering how catalytic pocket and residue volumes also set enzymatic metalloproteins apart. In the same year, Vornholt et al. [56] published an article where methods to systematically engineer artificial metalloenzymes (ArMs) for specific purposes were proposed, in which machine learning was used to predetermine function from amino acid sequence. The authors explored the performance of neural networks, SVM, and gradient boosting on five catalytic reactions and found that gradient boosting provided the best overall performance for predicting enzymatic activity. A combination of machine learning and a systematic screening approach was used to identify active variants, which increased the activity of engineered variants by up to 15-fold compared with the wild type. Overall, machine learning has revolutionized how metalloprotein functions are studied.

## 5. Protein Stability

Often, it is noteworthy to perform catalytic reactions in extreme environments beyond the tolerance of natural enzymes. This requires a purposeful design, such as the study by Liao et al. [53], which demonstrates how machine learning can contribute to improving the stability of protein structures under high heat.

As asserted before, one characteristic that makes metalloproteins suited for machine learning is the abundance of data collected about them. In 2019, Mazurenko et al. [57] published a perspective on incorporating machine learning to engineer enzymes, and enzyme stability was one of the many focal points. The authors highlighted how protein stability predictors (such as thermostability change or solubility change upon mutation) have the most plentiful data available for learning (see Table 1 of reference [57]). In general, protein stability makes for a suitable area of study all in itself, with its rich databases, such as ProTherm [58] among others.

In 2019, Montanucci et al. [59] used several machine learning methods, including SVMs and decision trees, to study the energetic stabilization/destabilization scale caused by a point mutation in metalloprotein structures. Stabilization/destabilization is defined by how the mutation changes the Gibbs free energy difference (ΔΔG), i.e., a negative change implies stabilization, and a positive change implies destabilization. Drawing from datasets like ProTherm, the upper limit of ΔΔG prediction was determined based on the uncertainty and spread of these values in the dataset, which reported multiple ΔΔG values of the same protein mutation under varying experimental conditions like pH and temperature. An upper bound of 0.8 for the Pearson correlation coefficient (between experimental and predicted data) was found [59]. Then, in 2020, along a similar line of motive to Mazurenko et al., Li et al. [60] presented ThermoNet, a convolutional neural network that takes the three-dimensional data of proteins with point mutations and trains it to predict if a particular mutation would stabilize or destabilize the structure, and whether it is benign or pathogenic. When a protein forms from its primary structure, it entails a Gibbs free energy change, which depends on the replacement of any amino acid within the sequence. The authors tested the program against the human p53 protein and myoglobin. For each mutation, there was a reverse mutation. While the model produced rather conservative predictions, it was generally unbiased and the predictions for mutation and reverse mutation followed a strong negative correlation.

ThermoNet foretells stabilizing and destabilizing effects with equal accuracy, whereas other ΔΔG predictive methods are biased towards predicting destabilizing effects. This bias purportedly arises from unbalanced training sets (consisting of primarily destabilizing mutations) and model overfitting. Figure 3 presents FoldX, a highly-used, non-specific ΔΔG predictor, as a comparison to ThermoNet using the ClinVar dataset as a benchmark. ClinVar contains ΔΔG values corresponding to benign and pathogenic missense variants. ThermoNet evidently outperforms ClinVar, with most predictions falling within the −5 kcal/moL to +5 kcal/moL experimentally-observed region. The discrepancy between FoldX values and ClinVar is increased for pathogenic variants, while ThermoNet maintains robust performance for either variant. With more analogous tools at hand, one may finally defer memorizing structural stability patterns to machines that can process that information with more efficiency and make quality predictions.

## 6. Inhibitor Design

Not all metalloprotein activities are beneficial. Under certain circumstances, their activities have to be inhibited when they are implicated in human diseases, like cancer. Moreover, since many microbes also rely on metalloproteins, inhibitors can be designed to combat microbial infections. Instead of using traditional methods like docking and molecular dynamics to predict inhibitors [61], using machine learning to screen inhibitors is a swifter and computationally inexpensive approach. One example is illustrated by Shi et al. [62], where the authors review work done to inhibit metallo-*β*-lactamases (MBLs) and destroy bacterial drug resistance. While surveying a variety of methods used to discover new molecules, the authors discussed the possibility of utilizing deep learning, given its accomplishments in chemical structure construction and predicting protein–ligand binding. This possibility has been realized in many similar works, as discussed below.

In 2018, Song et al. [63] published research on using random forests to identify inhibitors for matrix metalloproteases (MMPs). The inhibitors must selectively bind and take effect against carcinogenic MMPs while being innocuous towards those that counter the disease. With this in mind, the authors trained the model one MMP at a time for seven MMPs (MMP-2,3,7,8,9,13,14) to learn for each MMP the properties an inhibitor should possess. Judging from amino acid frequencies at different positions across 4000 peptide inhibitor samples, substantial new knowledge on what signature amino acid sequences should inhibitors of specific MMP types have and how specific the binding would be was added. Aside from Song et al., there was another work on a particular MMP by Li et al. [64]. Inhibitors for the MMP-12 enzyme were predicted using k-nearest neighbor (k-NN), random forest, C4.5 decision tree, and SVM. These models were trained against 90 inhibitors + 94 non-inhibitors before testing against 52 inhibitors + 47 non-inhibitors. Their performances vary, but all conform to very high standards, with accuracy ≳90%. In addition, a recursive feature elimination (RFE) capable of selecting choice features was appended so that the machine can classify inhibitory/non-inhibitory more effectively. Programs with RFE have higher performance than those without, but more crucially, it helped elect 36 essential features that help distinguish between MMP-12 inhibitors and non-inhibitors. The implication of these results goes far beyond one metalloenzyme family, since, theoretically, any enzyme that matches an extensive inhibitor database can become a subject of learning.

In later years, the scope of inhibitor studies has increased. For example, in 2021, Tinivella et al. [65] published research on inhibiting human carbon anhydrase (hCA), which strongly relates to cancer. Using algorithms (random forests, k-NN, SVM, NB, etc.) implemented in Python’s scikit-learn and drawing credible data from ChEMBL release 26, multiple models were trained to determine if a given molecule is an active inhibitor for any isoform of hCA, as well as how selective that molecule would be. The models include SVM, NB, and tree-based algorithms such as random forests, all displaying accuracy >70%, and their success significantly relates to a unique feature of the work. Traditionally, an hCA inhibitor is active when <20 nM of the enzyme remains active following inhibition and inactive when activity exceeds 100 nM. On the other hand, the authors decided to vary that threshold for each specific isoform and, as a result, considerable improvements were made. Another research study of a different theme was presented by Cañizares-Carmenate et al. [66]. Inhibitors for vasoactive metalloproteases to treat cardiovascular conditions were discovered. First, NB and multilayer perceptron (MLP) were tested as candidate models for a quantitative structure–activity relationship (QSAR) model and found the latter superior. Then, inhibitors predicted by QSAR were examined through docking experiments using thermolysin (TLN), a close bacterial homolog of human neprilysin and angiotensin-converting enzyme, both vasoactive metalloproteases. Ultimately, 18 possible chemicals with low binding free energy to TLN, a sign of effective binding, were identified, leading to an optimistic prediction of how the method will boost efficiency and save costs.

## 7. Cleavage Sites

The MMPs mentioned previously are a subgroup of metalloproteinases, which in turn are a group of proteases that cleave peptide substrates. All proteinases perform their action at an active site, a cleft structure responsible for catalysis. As proteases are among the most prominent enzyme families, they control a broad range of bioactivities and possess immense potential for biotechnological applications [67,68]. Therefore, it is essential to find these active cleavage sites and how they interact with substrates.

For metalloproteinases, much attention has focused on MMPs in recent years. In 2017, Wang et al. [68] conducted a study on predicting the cleavage site for multiple MMP types by using machine learning to make inferences about lesser-known MMPs using knowledge from better known ones (also known as transfer learning). Only for MMP-12 does transfer learning fail, while for MMP-3, it falls short for specificity alone. The other MMPs present transfer learning as irrevocably advantageous. Overall, the method demonstrates improvement over alternative computational methods mentioned, like PROSPER, Cascleave, etc. Nevertheless, as stated, the transfer learning algorithm will be subject to modification once more experimental data become available. Incidentally, Singh et al. [69] published an article in 2019, also on using transfer learning to predict where cleavage occurs in MMPs. Results produced by TrAdaBoost, an established boosting-based transfer learning algorithm, were compared to two of its variants called dynamic (D-) TrAdaBoost and multisource (M-) TrAdaBoost, and an SVM control group (see Table 2 of reference [69]). As aforementioned, transfer learning starts from better understood MMPs (source) to infer about less understood ones. For this purpose, shared enzymatic traits are assigned higher weights than non-shared. One flaw of TrAdaBoost is that these weights tend to converge after several boosting iterations, and the purpose of D-TrAdaBoost is to add a correction factor for this convergence [70]. In addition, M-TrAdaBoost tackles the possibility of poor equivalency of properties by allowing for multiple learning sources [71]. The results show that TrAdaBoost and its variants act optimally for different MMP types but do not consistently outperform SVM (and marginally when they do).

Further research has shown that one possible path to significantly enhancing performance relies on convolutional layers. Moreover, in 2019, Liu et al. [72] introduced DeepCalpain, a deep neural network (DNN) designed for cleavage site prediction in Ca^2+^-dependent metalloproteases called calpains. As a result, the network outperformed programs such as PoPS, GPS-CCD, and LabCaS much more significantly. The area under the curve (AUC) in Figure 4 is a measure of prediction reliability for classifiers, and DeepCalpain is the best performer. In addition, DeepCalpain can analyze mutated calpains present in various cancers and decide how the mutation affects cleaving target proteins. Using clinical data, it was determined that patients with no less than six mutations in calpain cleavage sites are significantly less likely to survive liver hepatocellular or head and neck squamous cell carcinoma. Another application of convolutional neural networks (CNNs) is in the analysis run by Li et al. [73] using the DeepCleave program on caspases and MMPs. DeepCleave was used to discern cleavage sites and the identity of substrates. Using three convolutional layers to build the CNN, the authors used kernels of three different sizes for the second and third layers. Following those layers, an attention layer automatically selects the most relevant generated features. These features are tunable, allowing a comparison to assess the performance of having different numbers of kernels and the presence and absence of an attention layer across three caspases (caspases-1,3,6) and MMPs (MMP-2,9,7). Results show that the attention layer impact performance the most and that at its best, DeepCleave rises to the standards of the most cutting-edge predictors, such as Cascleave and PROSPER. In brief, there is ample evidence to believe that algorithmic sophistication is key to making better predictions, the only issue being that the scope of the study has remained relatively limited. Future works should focus on expanding the categories of metalloproteins researched.

## 8. Conclusions

We provided an overview on the advances of machine learning applications in the field of metalloproteins, which is of great scientific interest due to their universality and versatility in design and function. On the whole, the machine learning procedure is to construct a suitable model that addresses the aim of study, train it using a set of (properly curated) data with features related to the property in question (such as structure, function, stability, etc.), validate the model, and make new predictions in order to test the model. Over the past two decades, this method has grown in reliability and popularity. Moreover, it has the potential to guide and readdress how mechanisms underlying metalloprotein function are understood fundamentally. For example, machine learning has provided new insights into how mutations affect structure and function, how diseases, such as cancer, relate to pathological alterations to metalloenzymes, and how treatments for such diseases can be customized. Overall, as machine learning advances and more data becomes widely available, significant strides will take place in understanding how metalloproteins influence biological processes. As discussed below, however, new challenges lay ahead.

## 9. Future Directions

Limitations persist for several points of interest. The most notable shortcoming is that most works remain on the level of supervised learning by drawing labeled data from established databases, with scant applications of unsupervised learning, such as generative models. While most studies focus on property prediction, and fewer studies relate to *de novo* design, the examples of designing artificial metalloenzymes (Section 4) and the discovery of new metalloprotease inhibitors (Section 6) show that machine learning is progressing along the right path. Nevertheless, future research should focus more on expanding the breadth and depth of experimental data applied to these predictive models. That being said, ensuring that the data is adequately curated is of utmost importance for model training, as the classic saying goes, “Garbage in, garbage out“.

Another insufficiency lies in the somewhat limited scope of metalloproteins studied. Studies investigating cleavage sites, for instance, focus predominantly on MMPs, and stability preponderantly implies under-heating. However, the human body is vulnerable to multiple factors that impact protein activity (pH level imbalance, heavy metal poisoning, etc.). Additionally, many essential metal-binding proteins come from myriad families, such as hemoglobin, as discussed above. Therefore, expanding the range of studies to other families would be meaningful.

Lastly, the exclusive focus on human metalloproteins constitutes an issue as well. As mentioned in the beginning, these proteins are ubiquitous and are crucial to viability across species. Understanding how photosynthetic metalloproteins are engineered, for example, could boost agricultural production. Moreover, knowing how drugs can be designed to inhibit metalloenzymatic actions in harmful microbes could help cure infections. As premature as this goal might be, it is worthwhile and profitable.

## Figures and Tables

**Figure 1 molecules-27-01277-f001:**
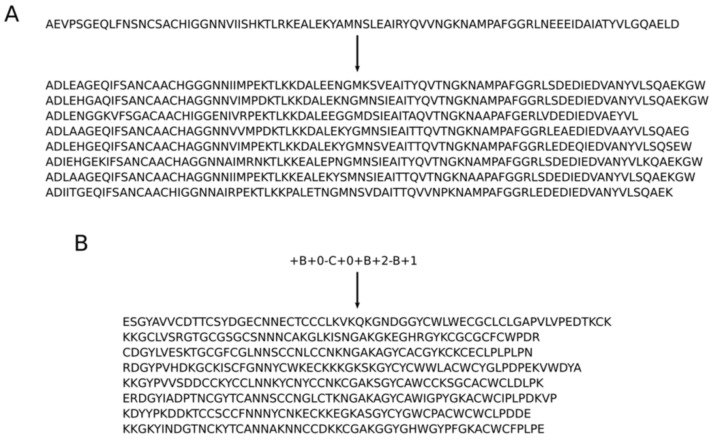
Amino acid sequences generated by CVAE as determined by (**A**) similarity to input sequence and (**B**) encoded protein topology that these sequences should conform to. This figure is reproduced with permission from reference [47], in accordance with a Creative Commons Attribution 4.0 International License (http://creativecommons.org/licenses/by/4.0/), accessed on 2 February 2022.

**Figure 2 molecules-27-01277-f002:**
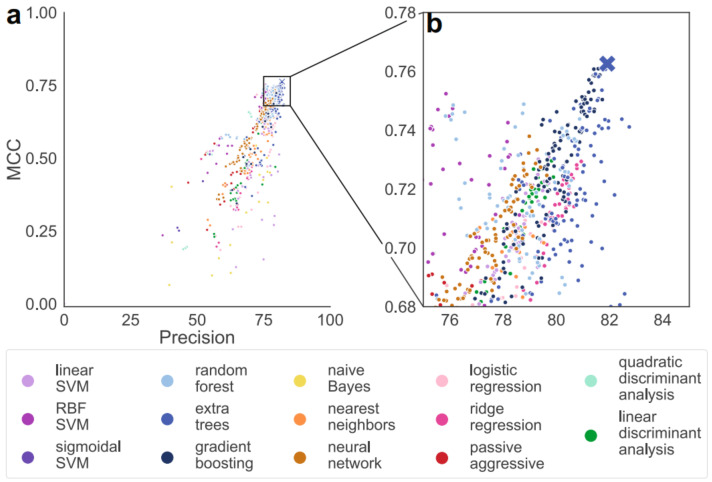
Comparison among candidate models according to precision and Matthews correlation coefficient (MCC). The cross represents the optimal model that maximizes both metrics, and is used by the authors for further tests. This figure is reproduced with permission from reference [12] in accordance with a Creative Commons Attribution 4.0 International License (http://creativecommons.org/licenses/by/4.0/), accessed on 2 February 2022.

**Figure 3 molecules-27-01277-f003:**
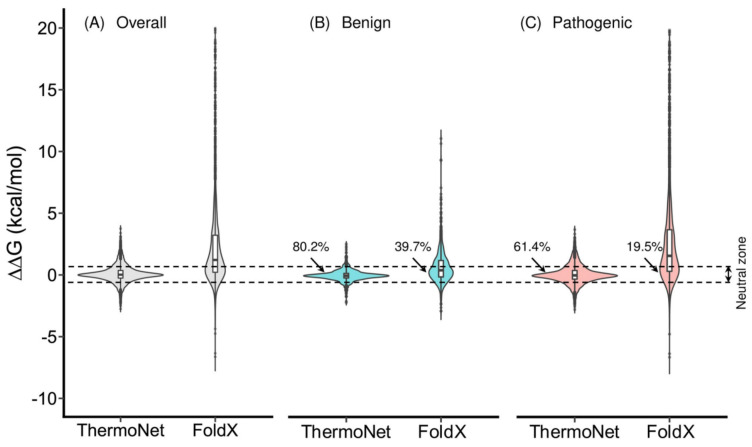
Comparison of the performance between ThermoNet and FoldX. This figure is reproduced with permission from reference [60], accessed on 2 February 2022.

**Figure 4 molecules-27-01277-f004:**
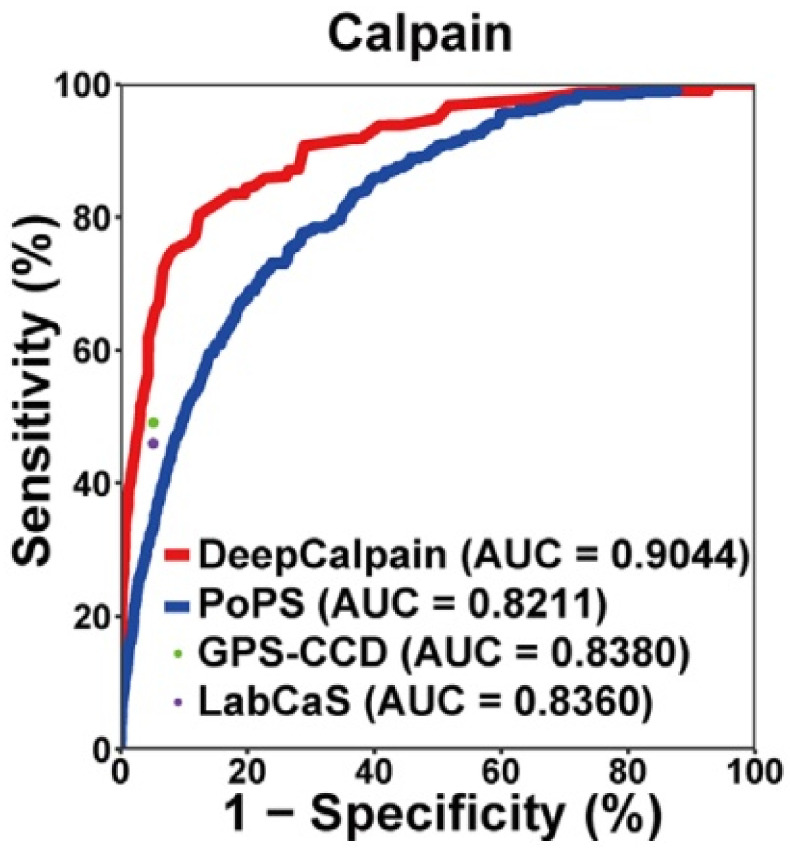
DeepCalpain compared with similar protease prediction programs according to AUC. This figure is reproduced with permission from reference [72], accessed on 2 February 2022. Copyright © 2019 Liu, Yu, Dong, Zhao, Liu, Zhang, Li, Du and Cheng.

## Data Availability

Not applicable.

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
