# Peer review of "Machine Learning Approaches for Metalloproteins"

_molecules, 2022, doi:10.3390/molecules27041277_

Round 1
Reviewer 1 Report
T
This publication deals with the application of machine learning methods in the study of the structure, activity and interaction with protein ligands from the families of metalloproteinases. The topic is very current and modern, but it is worth emphasizing how this approach may differ from other types of proteins, e.g. receptors and what are the difficulties (catalytic activity, presence of metals)
At the same time, the description lacked at least a brief mention of the classic molecular modeling methods used in this aspect (docking, molecular dynamics, QM calculations) and the justification why the ML approach is beneficial and sometimes necessary to apply (which may make that the reader understand the complexity of modeling approach)
At the same time, from the reader's point of view, it is worth including a short introduction to machine learning methods and a brief description of the various approaches used in this approach.
Author Response
1) Thank you for the comments. We included how this approach differs from other types of proteins and associated challenges (Line 77-81).
2) We have included a brief mention of the traditional molecular modeling methods used and the justification why the ML approach is beneficial and sometimes necessary to apply (Line 30-34, 38-40, Line 352-354). We also included how machine learning can complement traditional approaches (Line 153-156).
3) We have also included a short introduction about machine learning and the corresponding references for readers to learn more (Line 40-52).
Reviewer 2 Report
This is an interesting review paper describing the usage of machine learning techniques for the studies of metalloproteins. Indeed, the increase of the availability of structural information on metalloproteins results into a concomitant augmentation of structural analysis methods designed for the characterization of metal sites within proteins both from the structural and the functional points of view. The article has a very meticulous description of the machine-learning methods developed in this field. However, in order to make the review more complete, I would suggest to include into the review several important methodological milestones in the employment of machine learning for metalloproteins, which were not discussed: machine learning method FEATURE, a hybrid approach for recognizing calcium-binding sites in disordered regions based on the machine learning for structure-based site recognition. Although it could identify around 70% of crystallographically proven calcium-binding sites, it was an important step showing that the limited loop modeling combined with pattern matching algorithms could recover functions and propose putative conformations associated with these functions [https://doi.org/10.1186/1472-6807-9-72]; DeepMBS, predicting protein metal binding sites which was the first realization of deep learning idea for the problem of predicting metal binding site [https://doi.org/10.1109/MCSI.2017.13], Passerini’s MetalDetector v2.0: predicting the geometry of metal binding sites from protein sequence [https://doi.org/10.1093/nar/gkr365], MetalExplorer predicting metal-binding sites via a random forest algorithm [https://doi.org/10.2174/2468422806666160618091522].
Also, there are several very recent articles on the deep learning methodology employed for metalloproteins: loss of function mutations [https://doi.org/10.1038/s42256-019-0128-y], structures [https://doi.org/10.1002/pro.4074], [https://doi.org/10.26434/chemrxiv.13207967.v1], and a review [https://doi.org/10.1038/s41570-021-00339-5], which should also be taken into account.
Upon implementing these minor corrections, the manuscript will be publishable in Molecules.
Author Response
1) Thank you for the comments and suggested references.
2) We have included the requested references: FEATURE (Line 148 onwards), DeepMBS (Line 119 onwards), and MetalDetector v2.0 (Line 117 onwards).
3) We have included all of the reviewer's references on the deep learning methodology employed for metalloproteins in the review. We have included a discussion based on the loss of function mutations [https://doi.org/10.1038/s42256-019-0128-y], which is in turn based on the Koohi reference (line 219 onwards), as well as the article by Zhang (line 153 onwards).
Reviewer 3 Report
Comments on the manuscript ID1602590 by Yu et al. entitled “Machine Learning Approaches for Metalloproteins”:
The authors review machine learning in studying metalloprotein structures. Мetalloproteins are good subject for studying by machine learning approaches because there is a large amount of experimental data. The review article is well written, full of content material and beneficial information. The information is presented in a straightforward and understandable manner. Starting from simplified models of metalloproteins studied by DFT computations (2000), the authors review the using of various algorithms (random forests, neural networks, support vector machines, etc.) in studying various aspects of metalloproteins (structure, function, stability, ligand-binding interactions, and inhibitors). A future outlook and some important avenues for future research are also pointed out.
Author Response
Thank you for the comments!